# Evaluation of the Safety and Immunogenicity of a Multiple Epitope Polypeptide from Canine Distemper Virus (CDV) in Mice

**DOI:** 10.3390/vaccines12101140

**Published:** 2024-10-04

**Authors:** Santiago Rendon-Marin, Daniel-Santiago Rincón-Tabares, Jorge H. Tabares-Guevara, Natalia Arbeláez, Jorge E. Forero-Duarte, Francisco J. Díaz, Sara M. Robledo, Juan C. Hernandez, Julian Ruiz-Saenz

**Affiliations:** 1Grupo de Investigación en Ciencias Animales—GRICA, Facultad de Medicina Veterinaria y Zootecnia, Universidad Cooperativa de Colombia, Bucaramanga 680001, Colombia; santiago.rendonm@udea.edu.co; 2Grupo Infettare, Facultad de Medicina, Universidad Cooperativa de Colombia, Medellín 050001, Colombia; juankhernandez@gmail.com; 3Grupo Inmunovirología, Facultad de Medicina, Universidad de Antioquia, Medellín 050001, Colombia; dsantiago.rincon@udea.edu.co (D.-S.R.-T.); jorgetabare@gmail.com (J.H.T.-G.); francisco.diaz@udea.edu.co (F.J.D.); 4Grupo PECET, Facultad de Medicina, Universidad de Antioquia, Medellín 050001, Colombia; natyac182@gmail.com (N.A.); sara.rebledo@udea.edu.co (S.M.R.); 5Grupo de Investigación en Microbiología Ambiental, Escuela de Microbiología, Universidad de Antioquia, Medellín 050001, Colombia; jorge.forero@udea.edu.co

**Keywords:** multiepitope, vaccination, domestic dogs, canine distemper virus, immune response, safety

## Abstract

Background: *Morbillivirus canis* is the etiological agent of a highly contagious disease that affects diverse domestic and wild animals. Vaccination is considered the most suitable strategy for controlling CDV dissemination, transmission, and distemper disease. However, the emergence of new CDV strains has led to the need to update the current vaccine strategies employed to prevent CDV infection in domestic and wild animals. Currently, there is a lack of effective alternatives for wild animals. Diverse computational tools, especially peptide-based therapies, enable the development of new universal vaccines. Objective: The aim of this study was to evaluate the safety and humoral and cellular immune response of a new generation of vaccines based on CDV peptides as single-peptide mixtures or multiepitope CDV polypeptides in mice. Methods: Twenty-four BALB/c mice were subjected to a three-dose regimen for 28 days. Seroconversion was evaluated via ELISA, and cellular immune responses were evaluated via flow cytometry through activation-induced markers (AIMs). Results: Compared with the placebo, the peptide mixture and multiepitope CDV polypeptide were safe, and seroconversion was statistically significant in the multiepitope CDV polypeptide and commercial vaccine (CV) groups. The numbers of antigen-specific CD4+CD134+ and IFN-γ+ T cells, CD8+ T cells and TNF-α- and IL-6-producing cells were greater in the mice immunized with the multiepitope CDV polypeptide than in the control mice. Conclusion: This combined approach represents a potential step forward in developing new immunization candidates or enhancing current commercial vaccines to control CDV disease in domestic dogs and wild animals.

## 1. Introduction

*Morbillivirus canis*, also commonly known as canine distemper virus (CDV), is a member of the *Paramyxoviridae* family and *Morbillivirus* genus. It is the causative agent for canine distemper, a disease that is highly contagious and induces multiorgan disease in dogs and other carnivores [1]. It exhibits broad cell tropism affecting epithelial, lymphoid, and neurological cells, resulting in a systemic infection encompassing respiratory, digestive, urinary, lymphatic, cutaneous, skeletal, and central nervous system (CNS) manifestations [2]. CDV particles are often spherical, enveloped virions with a nonsegmented single negative-stranded RNA (ssRNA), similar to other members of the order *Mononegavirales*. The genome, spanning 15,690 nucleotides, encodes eight proteins, including H and F proteins [3], the main antigenic determinants of CDV [4]; peptides from the N, H, and F proteins have been found in greater abundance than other viral proteins within the major histocompatibility complex (MHC) molecules [5]. The host range of CDV primarily encompasses species within the order *Carnivora*, which belong to diverse families and, in lower proportions, other significant families from different orders, including *Artiodactyla*, *Primates*, *Rodentia*, and *Proboscidea* [6,7]. Considering the wide range of species infected by CDV, research has explored cross-species transmission among wild and domestic animals since lethal disease has spread to endangered species worldwide [8].

Vaccination is one of the most important strategies for preventing viral infections by eliciting both humoral and cellular immune responses. The common commercially available vaccines for CDV are modified live virus (MLV) vaccines based on strains, such as Onderstepoort, Snyder Hill, Convac, Rockborn, or CDV3 [9]. Nonetheless, these live attenuated vaccines have the potential to induce symptomatic disease and, in some cases, lead to mortality in certain susceptible species owing to their retained replicative capacity within vaccinated animals [10]. Moreover, the immunization of puppies harboring maternal CDV-neutralizing antibodies may prove inefficient, as these antibodies can reduce the efficacy of live-attenuated CDV vaccines [11]. Recombinant CDV vaccines have been created by incorporating the CDV F and H proteins into a canarypox virus vector [12]. These vaccines have shown to be safe across a range of susceptible species, including domestic dogs, European ferrets (*Mustela putorius furo*), giant pandas (*Ailuropoda melanoleuca*), fennec foxes, meerkats (*Suricata suricatta*), and Siberian polecats (*Mustela eversmanni*) [13,14,15,16]; however, there is a lack of information about efficacy in a wide range of animals. Although some subunit or innovative epitope-based vaccines have demonstrated the ability to elicit an adequate immune response to other viral agents in both in vitro and in vivo models [17,18,19], it remains imperative to explore alternative CDV vaccines for the wide range of natural CDV hosts. The prevalence of neutralizing antibodies has been studied in some wildlife animals [20,21]. There is significant concern about CDV transmission and dissemination control strategies for wild animals because some endangered animals can disappear without any intervention for CDV infection, such as vaccination [10,22].

The vaccine development process involves intricate, labor-intensive, and expensive in vivo and in vitro protocols during both preclinical and clinical study phases. Recently, advancements in computational biology have alleviated the reliance on in vitro experiments, facilitating effective in silico designs [23] since epitope-based vaccines have been reported to be promising immunization alternatives for comprehensive safety and immunogenicity [17].

Diverse computational tools have emerged as crucial components for the development of next-generation vaccines [24]. The focus on immunogenetics, immunogenomics, systems biology, immune profiling, and immunoinformatics has given rise to vaccinomics [25]. This interdisciplinary approach involves the comprehensive study of host–vector–pathogen molecular interactions and the identification of potential protective antigens, such as peptides derived from pathogen proteins [26]. One specific application within vaccinomics is multiepitope-based vaccines, which rely on the in silico prediction of immunogenic peptides from antigenically dominant pathogen proteins [27,28].

Different vaccine models have been promising for a wide range of viruses, including SARS-CoV-2, such as multiepitope-based vaccines (MEVs) based on viral antigenic determinants [29,30], Marburg, a multiepitope vaccine against structural proteins [17]; and HIV, a multiepitope peptide and the Nef protein with novel cell-penetrating peptides [31]. Moreover, epitope-based vaccines have been explored against various viral agents, such as hepatitis B, which has a multivalent core virus-like particle [32]; influenza A, which is based on conserved epitopes of hemagglutinin to develop a universal vaccine [33], hepatitis C, which explores a “multivalent scaffolding” approach with different epitopes [34], and Mayaro, which includes B and T-cell epitopes from five structural polyproteins [35]. Notably, peptide-based vaccines that include different epitopes do not involve infectious material, enable the practical insertion of different molecules to increase immunogenicity, can be prepared in lyophilized form for advantageous storage, pose no risk of virulence reversal, and can be designed to incorporate multiple antigenic determinants [36]. The versatility and safety profile of epitope-based vaccines make them a compelling choice for vaccine development, aligning with the evolving landscape of modern vaccinology. With respect to epitope delivery, carrier proteins or presentations in a multimeric format, such as virus-like particles (VLPs) or nanoparticles, can enhance immune responses by extending the epitope’s half-life, thereby reducing renal clearance and susceptibility to proteolytic degradation [37]. Thus, VLPs have been demonstrated to accurately deliver immunogenic epitopes such as canine parvovirus-like particles that carry major antigenic epitopes of a giant panda-derived CDV [38] and other morbilliviruses [39]. However, conventional adjuvants could enable the proposal of suitable vaccine options for animals [40].

To develop a new generation of safe vaccines on the basis of genetic and antigenic information on CDV linages circulating in domestic and wild animals, single peptides or multiepitope CDV polypeptides that were previously assessed in silico and in vitro [41] were evaluated in mice by exploring their safety and humoral and cellular immune responses. This approach enables the validation of the use of multiple immunogens, such as peptide mixture or multiepitope polypeptides, as potential CDV vaccines, which could be safe and highly immunogenic for domestic dogs and wildlife animals since there is a lack of new approved alternatives for protecting a wide range of animals threatened by CDV.

## 2. Materials and Methods

### 2.1. Ethical Approval

This study was approved by the ethics committee of the Universidad de Antioquia (Act No. 154, 8 August 2023). The authors also applied the three Rs principle and employed the ARRIVE guideline.

### 2.2. Peptides and Reagents

BIOMATIK (Wilmington, DE, USA) synthesized single peptides and polypeptides via standard solid-phase synthesis with purities >98% and characterized them via mass spectrometry (Table 1). Imject^®^ Alum (Thermo Scientific, Wilmington, DE, USA) was used as an adjuvant in the single-peptide mixture and polypeptide preparation to immunize the mice. A recombinant vaccine (Recombitek c3, Merial. Reg. ICA: No. 8966-BV) was used as a positive control for the CV group.

### 2.3. Animals, Clinical Signs, and Safety

A total of twenty-four 8-week-old wild-type (WT) BALB/c female mice (Charles River, Portage, MI, USA) were used for this study following 3R recommendations and an acceptable value (E) of 20 [42]. The mice were randomly divided into four experimental groups of 6 mice each: single-peptide mixture, multiepitope polypeptide, recombinant vaccine, and placebo. Clinical signs and the GRIMACE score, for assessing pain through the characterization of changes in five facial features or action units [43], were evaluated daily. Body weight was measured weekly as a clinical sign to determine potential endpoints. The inoculation site was evaluated daily to evaluate any indication of local reactions attributable to the vaccine through heat, pain, and swelling. Daily observation of vital signs, symptoms, behavior, and mortality was conducted to detect any adverse reactions to the vaccines.

### 2.4. Mouse Immunization and Sacrifice

The in vivo experiments were conducted within controlled, pathogen-free settings at the animal facility of the Universidad de Antioquia (Medellín, Colombia). The mice from all the groups were immunized subcutaneously, with a first dose on day 1 and two boosts on days 14 and 21 to increase immunogenicity [37,44]. A total of 100 μL of a 1:1 mixture of either a single-peptide mixture (20 nM of each peptide) or a multiepitope CDV polypeptide (25 nM) diluted in 0.9% saline solution and Imject^®^ Alum (Thermo Scientific, Wilmington, DE, USA) was used for the single-peptide mixture and multiepitope CDV polypeptide groups; 100 μL of recombinant vaccine or 0.9% saline solution was used for the recombinant vaccine and placebo groups, respectively. All vaccinated mice were immunized with vaccines from the same batch. On day 28, all the mice were euthanized via an intraperitoneal overdose of ketamine/xylazine (100/10 mg/kg).

### 2.5. Splenocyte Isolation and Stimulation

Once euthanized, whole mouse spleens were extracted under sterile conditions, added to transport medium (RPMI supplemented with 5% penicillin–streptomycin), and covered with ice for splenocyte isolation. Briefly, the spleens were mechanically disrupted in a cell strainer (BD Biosciences, San Jose, CA, USA) and resuspended in RPMI supplemented with 10% fetal bovine serum (FBS) (Gibco, Grand Island, NY, USA). The cell suspensions were washed three times and centrifuged for 5 min at 1800 rpm. Then, red blood cell (RBC) lysis buffer was used (eBioscience, San Diego, CA, USA) for five minutes, after which the cells were washed with 1X PBS and centrifuged for five minutes at 1800 rpm. The obtained cells were counted and seeded in 96-well plates (1 × 10^6^ cells/well) and were stimulated for 24 h with either 25 nM polypeptide or 8 μg/mL phytohemagglutinin-PHA (Sigma–Aldrich, St. Louis, MO, USA). Nonstimulated cells were also used as a negative control for each mouse.

### 2.6. Evaluation of the Splenocyte Population by Flow Cytometry

Splenocyte populations and activation markers were assessed using flow cytometry after 24 h of polypeptide stimulation. Prior to staining, after 8 h of culture, 6 mg/mL brefeldin A and 2 mM monensin (Thermo Scientific, Wilmington, DE, USA) were added to the cell culture and incubated at 37 °C and 5% CO_2_. The cells were subsequently washed and stained for 30 min in the dark with a cocktail of the following antibodies: a FITC-conjugated monoclonal antibody against CD3 (clone: 145-2C11), a V500-conjugated monoclonal antibody against CD4 (clone: RM4.5), a PerCP-Cy5.5-conjugated monoclonal antibody against CD8 (clone: 53-6.7), a BV650-conjugated monoclonal antibody against CD25 (clone: PC61), and a BV421-conjugated monoclonal antibody against CD134 (clone: OX-86) (all antibodies from BD Biosciences, San Jose, CA, USA). The cells were fixed with mouse Foxp3 buffer (BD Biosciences, San Jose, CA, USA) according to the manufacturer’s guidelines and labeled with a PE-Cy-7-conjugated monoclonal antibody against IFN-γ (clone: XMG1.2). A final wash was carried out, and cell populations were acquired via an LS Fortessa (BD Biosciences, San Jose, CA, USA). The data were analyzed via FlowJo version 10.5.3 (FlowJo, LLC, Ashland, OR, USA) and normalized to those of nonstimulated cells as a negative control for each mouse.

### 2.7. Cytokine Quantification by Cytometric Bead Assay (CBA)

The splenocyte culture supernatants were collected from independent wells from the splenocytes used for flow cytometry and stored at −80 °C until use. The supernatants were subsequently thawed at 4 °C before the CBA was applied. A Mouse Th1/Th2/Th17 Cytokine Kit, which allows interleukin (IL)-2, IL-4, IL-6, IFN-γ, TNF-α, IL-17A, and IL-10 protein determination, was used (BD Biosciences, San Jose, CA, USA). A CBA was carried out according to the manufacturer’s instructions. The data were acquired using a CytoFLEX (BC Life Biosciences, Brea, CA, USA). The cytokine standards were serially diluted to construct calibration curves, which were necessary to determine the protein concentrations of the mouse samples. Individual cytokine concentrations are indicated by their fluorescence intensities converted to concentrations (pg/mL) via FlowJo version 10.5.3 (FlowJo, LLC, Ashland, OR, USA).

### 2.8. In-House ELISA

Flat-bottomed ELISA plates (Thermo Fisher Scientific, Wilmington, DE, USA) were coated overnight at 4 °C with 100 ng per well of multiepitope CDV polypeptide employed as an immunogen diluted in 0.5 M carbonate-bicarbonate buffer. The plates were washed five times with PBS containing 0.05% Tween (PBST), blocked in 1% bovine serum albumin (BSA) solution for 1 h at 37 °C, and washed five times with PBST. Mouse serum samples were prepared at a 1:200 dilution in PBS supplemented with 10% fetal bovine serum (FBS), 100 μL per well, and incubated for 2 h at 37 °C. The wells were washed five times with PBST. Then, 100 μL of conjugated goat anti-mouse IgG (H + L)-HRP (CAT1706516-Bio-Rad) at a dilution of 1:2000 in PBS supplemented with 10% FBS was added to each well, and the mixture was incubated for 1 h at 37 °C. After seven washes with PBST, 100 mL of 3,3′,5,5′-tetramethylbenzidine (TMB) was added to each well for 15 min for color development, and the reaction was stopped with 50 μL of 2 N HCL. The absorbance was measured at 450 nm with a microplate reader (Thermo Fisher Scientific, Wilmington, DE, USA). The optical density (OD) values were calculated by subtracting control values (wells without serum) to all samples. The average specific IgG concentration was determined by 2 independent experiments for each mouse serum sample for each experimental group.

### 2.9. Statistical Analysis

Descriptive statistics are shown as the means and standard deviations or medians and interquartile ranges (IQRs), depending on normality. The data were analyzed using an unpaired Student’s *t*-test or the Mann–Whitney U test, depending on the normality assumption of the Shapiro–Wilk test. Statistical significance was considered at * *p* < 0.05, ** *p* < 0.01, *** *p* < 0.001 and **** *p* < 0.0001, and statistical significance was determined with GraphPad Prism Software (La Jolla, San Diego, CA, USA, version 10.1).

## 3. Results

### 3.1. Clinical Signs, Weight Loss and Clinical Following

To determine the safety of the CDV peptide mixture and the multiepitope CDV polypeptide, the immunized mice were observed daily after the complete immunization scheme (Figure 1a); none of the mice exhibited abnormal behavior or clinical symptoms of toxicity on the basis of the GRIMACE scale (facial actions such as orbital tightening, nose bulge, cheek budge, ear position, and whisker change), which was 0 throughout the entire study. Weekly weight was measured as an adverse effect of the CDV peptide mixture, the multiepitope CDV polypeptide and commercial vaccine immunization. For all the groups, the weight of the mice increased. None of them presented a weight loss greater than 10% (Figure 1b–e). Therefore, immunization with a CDV peptide mixture, a multiepitope CDV polypeptide and a commercial vaccine (CV) was safe in mice.

### 3.2. Mice Immunized with the Multiepitope CDV Polypeptide Presented Increased Antigen-Specific IgG

An in-house ELISA was performed to assess antibody production in all the mouse groups. After euthanasia, total blood was collected, and the serum was separated to measure the presence of serum antigen-specific IgG with the CDV polypeptide as an antigen for coating. As shown in Figure 2, immunization with the CDV peptide mixture did not significantly increase antibody levels, as did immunization with the placebo. However, when polypeptides were employed, there was a statistically significant increase in antibody production compared with that in the placebo group (*p* = 0.0411). Moreover, polypeptide-specific IgG production was significantly greater in the CV group than in the placebo group (*p* = 0.0036). Therefore, compared to placebo, immunization with either the multiepitope CDV polypeptide or CV increased antibody production and induced more IgG than immunization with the CDV peptide mixture. However, CV-immunized mice had the highest multiepitope CDV polypeptide-specific IgG production.

### 3.3. The Multiepitope CDV Polypeptide Induces a Cellular Immune Response

To establish whether immunization with either the peptide mixture or the polypeptide induces a cellular immune response, flow cytometry assays were carried out to measure vaccine-responsive CD4+ and CD8+ T cells in splenocytes. A representative gating strategy for CD4+ T cells and activation AIMs is shown in Figure 3a. These results revealed a trend toward an increase in the percentage of antigen-specific CD25+ CD4+ T cells in the CDV polypeptide group compared with those in the placebo, peptide mixture and CV groups (Figure 3b). On the other hand, the percentage of antigen-specific CD134+ (Ox40) CD4+ T cells was greater than that in the placebo group (*p* = 0.0306, Figure 3c). Although the percentage of double-positive antigen-specific CD25+ CD134+ CD4+ T cells tended to be greater in the multiepitope CDV polypeptide group than in the placebo group, the difference was not statistically significant (Figure 3d). The percentage of CD4+ T cells that were IFN-γ+ was significantly different (*p* = 0.0398) between the multiepitope CDV polypeptide group and the placebo group, as shown in Figure 3e. Consequently, the cell-mediated immune response induced by specific CD4+ T cells increased when the multiepitope CDV polypeptide was used for immunization in mice compared with that in the placebo group, and the CV group did not exhibit a considerable CD4+ T-cell immune response.

For CD8+ T cells, a representative gating strategy for antigen-specific CD3+ CD8+ T cells identified by flow cytometry analysis is shown in Figure 4a. BALB/c mice immunized with the multiepitope CDV polypeptide produced more antigen-specific CD8+ T cells than did those in the placebo, peptide mixture and CV groups, but the differences were not statistically significant (Figure 4b). Therefore, immunization with the multiepitope CDV polypeptide could induce a slight increase in specific CD8+ T cells, indicating the potential for the development of an essential aspect of the cell-mediated immune response and no appreciable increase in antigen-specific CD8+ T cells in mice immunized with CV.

### 3.4. Cytokine Production in Splenocytes Stimulated with the Multiepitope CDV Polypeptide

To measure cytokine production in splenocyte cultures, a cytokine bead array was performed to determine cytokine-producing Th1/Th2/Th17 CD4+ populations as activation and differentiation markers. As shown in Figure 5, for CDV peptide mixture immunization in mice, no statistically significant differences were observed in the amounts of any cytokine compared with those in the placebo group. On the other hand, TNF-α (*p* = 0.0281) and IL-6 (*p* = 0.0152) were significantly increased in the multiepitope CDV polypeptide group (Figure 5d,g). Moreover, there were significant differences in the concentrations of INF-γ, IL-2, IL-6, IL-17 (*p* < 0.01), and TNF-α (*p* < 0.05) after CV immunization compared to the placebo group. Notably, no significant differences in the IL-4 or IL-10 concentrations were detected among the evaluated groups. Taken together, these results regarding cytokine expression in splenocytes indicate that multiepitope CDV polypeptide immunization induces the proinflammatory cytokines IL-6 and TNF-α, which are important for CD4+ T cells; however, CV can induce cytokine profiles different from those of Th1/Th2/Th17 CD4+ cells.

## 4. Discussion

CDV vaccine development has been explored by diverse researchers worldwide in vivo in different species, such as domestic dogs, BALB/c mice, minks, and ferrets, among others, on the basis of recombinant viruses in the backbone of adenoviruses and canarypox viruses [45,46,47,48], chimeric measles virus-expressing CDV proteins [49], DNA vaccines based on the expression of CDV antigenic determinants [50,51,52], pure H and F CDV proteins as antigens [53], recombinant mouse adenovirus 1 (MAV-1) expressing CDV antigens [54], and novel bacterium-like particle-based vaccines displaying canine distemper virus antigens [9]. Nevertheless, CDV peptide-based vaccines have not yet been investigated.

In this study, we employed a three-dose CDV peptide-based vaccine evaluated in silico and in vitro previously [41] in mice to demonstrate safety and immunogenicity (Figure 1a). Common approaches in vaccine development typically employ entire microorganisms, which could lead to inadvertent exposure in susceptible wildlife animals [55]. Peptide vaccines, comprising short immunogenic peptide fragments, could present a viable solution to this issue by eliciting potent and targeted immune responses while mitigating the risk of safety [36]. Accordingly, immunization of mice with the CDV peptide mixture and the multiepitope polypeptide resulted in an excellent safety profile, as measured through weight percentage changes and the GRIMACE scale (Figure 1b–e).

To generate a robust immune reaction, any vaccine alternative must possess the capacity to stimulate innate and adaptative immune responses. The humoral immune response is considered the gold standard for evaluating vaccine candidate efficacy against viral pathogens [56]. In mice, several approaches have demonstrated the importance of measuring specific IgG production as an accumulative effect of immunization for viral pathogens such as CDV [9] and considering the importance of universal vaccines for CDV. Our results indicated that antigen-specific IgG levels measured after multiepitope CDV polypeptide immunization were greater than those in the control group but lower than those in the CV group, as shown in Figure 2. This finding is consistent with previous studies on CDV vaccine development in which specific IgG antibodies increased after immunization with new recombinant or DNA-based vaccines [48,50], demonstrating the importance of the humoral immune response in effective vaccine candidates.

For agents belonging to the *Morbillivirus* genus, such as CDV, both cytotoxic T lymphocyte activity specific to the H protein [57] and helper T-cell epitopes from the F protein [58] have been widely described. Cellular immunity mediated by CD4+ and CD8+ T cells is indispensable for protection against CDV infection and disease [59]. The AIM assay enables the identification of antigen-specific T cells on the basis of the upregulated expression of activation markers after antigen restimulation [60]. Although this technique has not been widely employed in mice, some studies have used this approach in lymphocytic choriomeningitis virus studies to identify T cells that upregulate AIMs, such as CD134 and CD25, after cell culture with specific antigens, allowing for the quantification of murine antigen-specific CD4+ T cells [61]. Moreover, AIMs are considered especially advantageous for discerning antigen-specific T follicular helper (Tfh) cells, which constitute a CD4+ T-cell subset crucial for supporting B cells [62]. For example, this technique has been employed to detect hemagglutinin-specific Tfh cells by observing the increased expression of CD25, CD134, and CD154 after IAV infection or influenza A virus hemagglutinin immunization in C57BL/6 mice [63]. Here, we demonstrated an increase in the number of antigen-specific CD4+ T cells, such as CD25-, CD134- or double-positive cells, with increasing AIMs after stimulation with the multiepitope CDV polypeptide (Figure 3) and after immunization with three doses. Hence, AIMs have emerged as an important strategy for preclinical investigations into murine vaccines, facilitating the assessment of the proportional abundance of vaccine antigen-specific cells [61]. On the other hand, antigen-specific CD8+ T cells were also detected (Figure 4), which is consistent with recent studies based on bacterium-like particle-based vaccines displaying CDV antigens, where specific CD8+ T cells were also increased in mice and dogs [9]. There were more specific IFN-γ-secreting CD4+ T cells (Figure 3e), indicating the potential stimulation of cytotoxic T-cell activity through the Th1 CD4+ T-cell subset after multiepitope CDV polypeptide stimulation of mouse splenocytes.

Cytokine production is associated with the polarization and production of specific CD4+ T-cell subsets [64]. Our results revealed a statistically significant difference in TNF-α and IL-6 production between splenocytes from the multiepitope CDV polypeptide mouse group and those from the placebo group (Figure 5d,e). TNF-α is a proinflammatory cytokine that is important for naive T-cell activation and proliferation [65]. Moreover, IL-6 enhances vaccine responses by promoting Tfh cells [66] and subsequent antibody production [67]. IL-6 triggers the activation of transcription factors, specifically STAT3, via Janus kinases (JAKs) and CCAAT/enhancer-binding proteins (C/EBPs) through the ras-ERK mitogen-activated protein kinase (MAPK) cascade. The activation of STAT3 leads to the upregulation of c-maf expression, whereas C/EBP facilitates the upregulation of NFATc2. The transcription factors c-maf and NFATc2 may collaboratively facilitate the differentiation of CD4+ Th2 or Tfh cells [68,69,70]. Thus, the production of IL-6 after stimulation with the multiepitope CDV polypeptide could assist the relevant signaling interplay of Tfh cells, a subset of CD4+ cells that produce antibodies by B cells. When CV was used, different cytokines from CD4+ T cells were detected, indicating the potential proinflammatory profile of the recombinant vaccines, as previously reported [48].

The production of cytokines that help antigen-specific Tfh cells, a CD4+ T-cell subset and antibodies increased in the multiepitope CDV polypeptide and CV mouse groups, but the peptide mixture group exhibited no humoral response (Figure 2, Figure 3 and Figure 5). There are several factors that could explain why the single-peptide mixture failed, whereas the multiepitope CDV polypeptide generated a humoral and cellular immune response. The polypeptide was designed with amino acid linkers to facilitate processing for presentation on MHC-I and MHC-II molecules. However, for individual peptides, there is a possibility of degradation before presentation [36]. The potential to be presented as exogenous antigens, displacing already loaded peptides on MHC molecules, decreases the likelihood of presenting these peptides to generate an immune response in antigen-presenting cells compared with the processing and presentation mechanism in a conventional alternative [71]. The single-peptide mixture could be complex enough to guarantee the bioavailability of each peptide in a single-peptide mixture compared with a multiepitope polypeptide, a larger molecule that includes all immunogenic epitopes [36] and usage of other carrying molecules as lipids [72] and nanoparticles [73] to improve peptides delivery and immunogenicity.

The biological events associated with a protective adaptative immune response against CDV epitopes oriented toward a specific antigen have been reported for other vaccine candidates with different immunogens, such as bacteria-like particles [9]. Although an adjuvant that potentiates the humoral immune response was employed, the CV vaccine induced more specific IgG than both the peptide mixture and the multiepitope CDV polypeptide. Moreover, increased numbers of CDV-specific IFN-γ-secreting CD4+ T cells were observed in multiepitope CDV polypeptide-immunized mice, indicating that Th1 CD4+ T cells are also important for mediating cytotoxic T lymphocyte activity since the number of CD8+ T cells tended to increase in the multiepitope CDV polypeptide group compared with the placebo group (Figure 4). Several reasons enable the investigation of peptide-based vaccines, such as their absence of infectious agents, convenient practical integration of molecules to increase immunogenicity, easy storage, and the incorporation of multiple antigenic elements [36]. However, several limitations may contribute to the low immunogenicity of peptide-based vaccines that must be overcome, including the diversity of MHC molecules in antigen-presenting cells, proper entry of peptide vaccines into the MHC pathways, bioavailability, and immunogen concentrations [36], as demonstrated in this research. Moreover, further studies must be carried out to determine the long-term immune response generated by the multiepitope CDV polypeptide and evaluate new-generation adjuvants, such as CpG and lipid nanoparticles, that work as patron pattern recognition receptor agonists [40].

The canine distemper virus has demonstrated the capacity to overwhelm a widening spectrum of hosts, which could be considered a considerable obstacle to controlling and eradicating this disease. CDV can have severe effects on numerous endangered species, such as black-footed ferret (*Mustela nigripes*), Santa Catalina Island fox (*Urocyon littoralis catalinae*), African wild dog (*Lycaon pictus*), and Caspian seal (*Pusa caspica*), and can contribute to its decline or near extinction [74,75,76,77,78]. Additionally, outbreaks have been reported within captive breeding facilities that house endangered African wild dogs and threatened giant pandas (*Ailuropoda melanoleuca*) [74,79,80]. Moreover, large felids have led to CDV disease outbreaks and mortalities across various species of the *Panthera* and *Lynx* genera [76,81]. Currently, CDV remains a formidable threat to the Amur leopard (*Panthera pardus orientalis*), Javan leopard (*Panthera pardus melas*), Amur tiger (*Panthera tigris altaica*), and Asiatic lion (*Panthera leo persica*), all of which are endangered subspecies [82,83]. MLV and recombinant vaccines employing canarypox vectors against CDV have been utilized in commercial applications for carnivore protection worldwide [84]. Although the canarypox-vectored CDV vaccine was endorsed for all susceptible species by the American Association of Zoo Veterinarians’ Distemper Vaccine Subcommittee, there are still some issues to overcome, such as safety and availability in several countries, in addition to the notable individual and interspecies variability in response to this vaccine for each species. On the other hand, MLVs could represent a risk of severe disease and fatality in highly susceptible species since safety has been demonstrated only in domestic dogs and some animals, such as ferrets and African wild dogs [10,85], but there is still a vast array of species that could be at risk with commercially available vaccines; however, ethical limitations to studying vaccines for wildlife animals explain not only the low amount of studies reported in a previous scoping review [86] but also the imperative need of new studies focusing on CDV vaccines for wildlife animals. As a perspective, not only studies in the vast array of affected animals for CDV but also comparisons with other commercial vaccines, such as MLV, must be performed since several studies of CDV vaccine safety and immunogenicity in domestic and wild animals are under experimental phases [86].

Thus, universal vaccines based on noninfectious therapies and new-generation vaccines have arisen as safe alternatives in wild animals. In CDV-susceptible animals for which the safety and efficacy of current vaccines have not been demonstrated, peptide-based vaccines, especially multiepitope CDV polypeptide immunogens, in spite of its limitations, can be used [87,88], as evaluated in this study through a combined in silico, in vitro and in vivo approach. This could be a safe and effective alternative for CDV disease control and prevention, even as a booster in vaccinated animals with commercially available recombinant vaccines.

## 5. Conclusions

A CDV peptide-based vaccine was evaluated, either as a single-peptide mixture or a multiepitope CDV polypeptide. One initial immunization with two boosts in an immunization scheme within 28 days in mice induced both humoral and cellular immune responses when the multiepitope CDV polypeptide was employed. The immunogenic multiepitope polypeptide was formulated on the basis of linear B cells, cytotoxic T lymphocytes, and helper T lymphocyte epitopes previously reported [41]. After mice splenocyte stimulation with multiepitope CDV polypeptide, antigen-specific CD4+ T cells were identified, indicating a specific immune response to the multiepitope CDV polypeptide. Thus, the development of a multiepitope CDV polypeptide has become a promising strategy against viral infections, such as CDV or a potential booster for current commercially available recombinant vaccines, considering the response of the CV group to multiepitope CDV polypeptides. Moreover, improving peptide immunogenicity, using new-generation adjuvants, and exploring higher concentrations of peptides while considering their safe profile are imperative. It is essential to acknowledge that this preliminary-designed vaccine is not exhaustive and requires further in vivo experiments in target species, such as domestic dogs and endangered wildlife animals, to comprehensively assess its effectiveness, including virus challenge. However, our approach represents a considerable step forward in developing a new immunization candidate or alternative for controlling CDV disease and dissemination in domestic dogs and wildlife.

## Figures and Tables

**Figure 1 vaccines-12-01140-f001:**
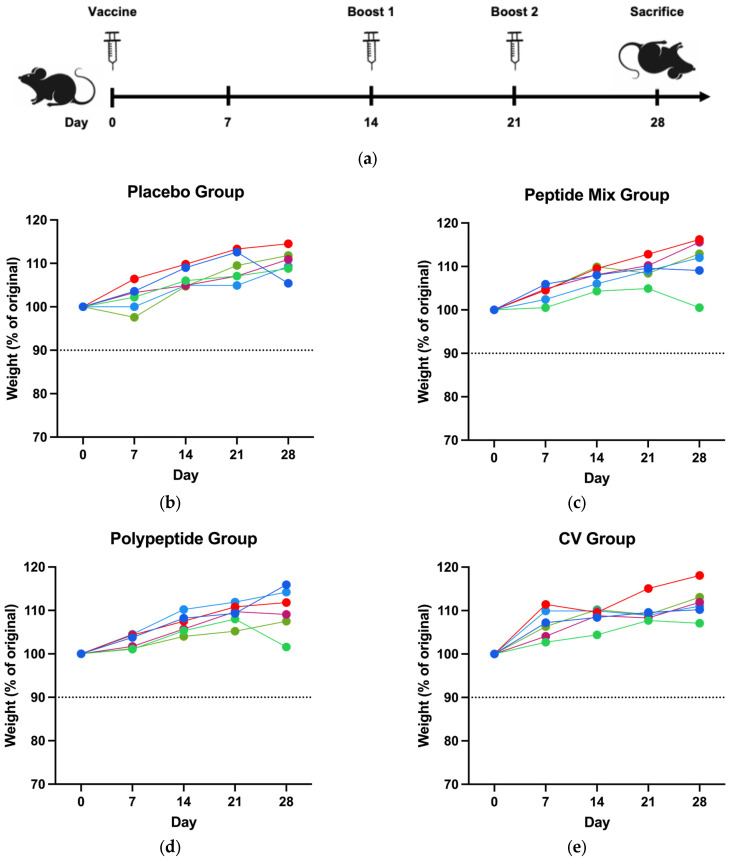
Immunization scheme and clinical signs. (**a**) Mouse immunization scheme with one vaccination and two boosts. The weight loss of the mice was monitored for all assay days until they reached sacrifice for the (**b**) placebo, (**c**) peptide mixture, (**d**) multiepitope CDV polypeptide, and (**e**) CV groups (90% in the dotted line represents the threshold of body loss). One single line corresponds to each mouse in all experimental groups.

**Figure 2 vaccines-12-01140-f002:**
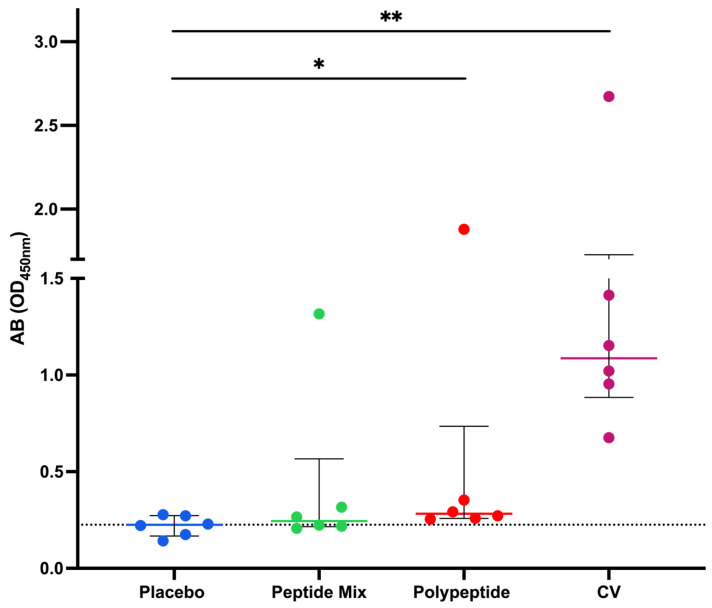
Immunization with the CDV polypeptide induced an increase in the level of antigen-specific IgG. Mice were immunized with peptide mixture or polypeptide. Antibodies were detected via ELISA employing the polypeptide as a coating antigen. The data are expressed as the means of two independent experiments for each mouse serum after the background absorbance was subtracted, and the medians ± IQRs of all the mice are also reported. Statistical evaluations were performed with an unpaired Student’s *t* test or the Mann–Whitney U test. * *p* < 0.05 and ** *p* < 0.01.

**Figure 3 vaccines-12-01140-f003:**
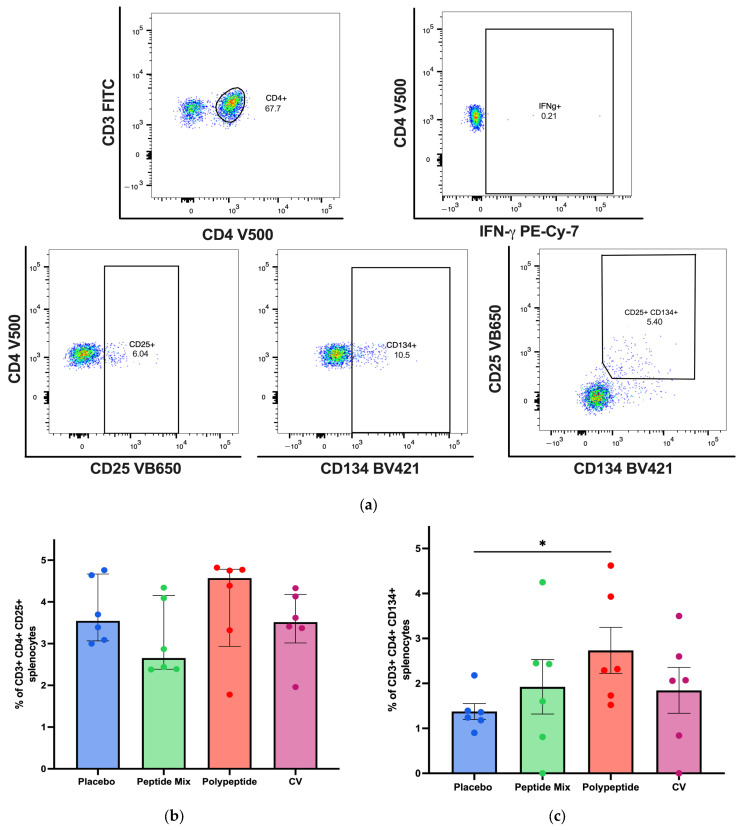
Specific cellular immune response induced in immunized BALB/c mice for CD4+ T-cell populations in splenocytes by flow cytometry. (**a**) A representative gating strategy for antigen-specific CD4+ T cells was evaluated in splenocytes from AIM patients (CD25+ CD134+ and double positive), and IFN-γ-producing cells were selected from the CD3+ cell population. (**b**) Percentage of antigen-specific CD4+ CD25+ T cells. (**c**) Percentage of antigen-specific CD4+ CD134+ T cells. (**d**) Percentage of antigen-specific double-positive CD4+ CD25+ CD134+ T cells. (**e**) Antigen-specific CD4+ T cells producing IFN-γ in each mouse group. All the cell cultures were evaluated for 24 h and stimulated with the multiepitope CDV polypeptide. The percentages were normalized to those of nonstimulated cells for each condition. The data are expressed as the median ± IQR. Statistical evaluations were performed with unpaired Student’s *t*-test or the Mann–Whitney U test. * *p* < 0.05.

**Figure 4 vaccines-12-01140-f004:**
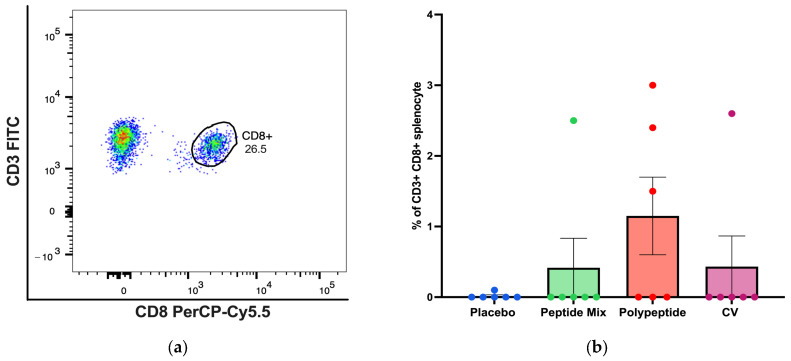
Specific cellular immune response induced in immunized BALB/c mice for CD8+ T-cell populations in splenocytes via flow cytometry. (**a**) A representative gating strategy for antigen-specific CD8+ T cells evaluated in splenocytes selected from the CD3+ cell population. (**b**) Percentage of antigen-specific CD3+ CD8+ T cells. All the cell cultures were evaluated after 24 h of culture and stimulation with the multiepitope CDV polypeptide. The percentages were normalized to those of nonstimulated cells for each condition. The data are expressed as the median ± IQR.

**Figure 5 vaccines-12-01140-f005:**
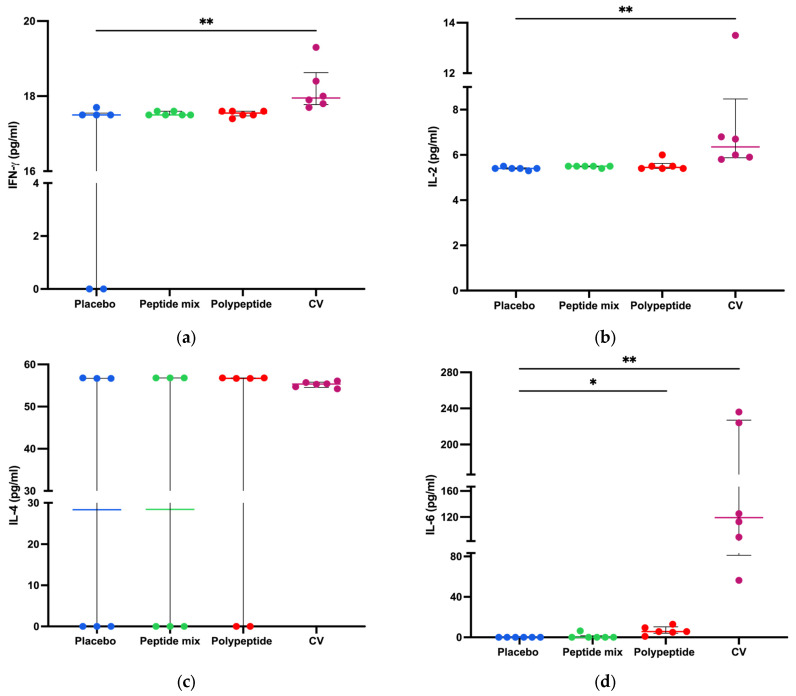
Splenocyte cytokine production in all immunized mice. Cytokine concentrations in CD4+ Th1/Th2/Th17 cells in the supernatants of 8 h splenocyte cultures stimulated with the multiepitope CDV polypeptide (pg/mL) measured via CBA. (**a**) IFN-γ; (**b**) IL-2; (**c**) IL-4; (**d**) IL-6; (**e**) IL-10; (**f**) IL-17; and (**g**) TNF-α. The data are expressed as the mean ± standard error of the mean (SEM) or median and interquartile range. Statistical evaluations were performed with unpaired Student’s *t* test or the Mann–Whitney U test. * *p* < 0.05 and ** *p* < 0.01.

**Table 1 vaccines-12-01140-t001:** CDV single peptides and multiepitope polypeptides.

Peptide	Sequence	Protein	Initial Position	Length (AA)	Purity (%)	Concentration (nM) **
**P1**	QVIDVLTPLFK	H	97	11	98.21	20
**P2**	VENLVRIRF	F	316	9	98.27	20
**P3**	LKLLRYYTE	H	592	9	98.45	20
**P4**	PPYLLFVLLILLV	H	32	13	98.12	20
**P5**	KAQIHWNNL	F	134	9	98.72	20
**Poly ***	QVIDVLTPLFK**AAY**LKLLRYYTE**GPGPG**VENLVRIRF**GPGPG**PPYLLFVLLILLV**KK**KAQIHWNNL	NA	NA	66	98.65	25

* Polypeptides were constructed with single peptides (P1, P2, P3, P4, and P5) linked with small amino acid sequences or linkers marked in bold [17]. ** The CDV peptide mixture was made into an equimolar solution. Each peptide was 20 nM in concentration. These concentrations refer to the final amount of peptides in the immunogenic preparation for mice.

## Data Availability

All the data are presented in the paper.

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
