# Peer review of "Evaluation of the Safety and Immunogenicity of a Multiple Epitope Polypeptide from Canine Distemper Virus (CDV) in Mice"

_vaccines, 2024, doi:10.3390/vaccines12101140_

Round 1

Reviewer 1 Report (Previous Reviewer 1)

Comments and Suggestions for Authors

The manuscript titled “Evaluation of the safety and immunogenicity of a multiple 2 epitope polypeptide from canine distemper virus (CDV) in mice” addresses a critical issue in veterinary medicine by focusing on the development and evaluation of a novel multiepitope vaccine against Canine Distemper Virus (CDV). Given the broad host range of CDV and its impact on both domestic and wild animal populations, this research is highly relevant and contributes to the ongoing efforts to control and prevent CDV infections. The study employs a cutting-edge vaccinomic approach, leveraging computational biology and immunoinformatics to design a multiepitope CDV vaccine. This approach is innovative as it allows for the prediction and selection of immunogenic peptides, potentially enhancing vaccine efficacy and safety. The authors have conducted a thorough evaluation of the vaccine's safety and immunogenicity in mice, including both humoral and cellular immune responses. The use of a well-established animal model (BALB/c mice) and the inclusion of both single peptide mixtures and a multiepitope polypeptide provide a comprehensive analysis of the vaccine's performance. The manuscript provides detailed descriptions of the experimental procedures, including peptide synthesis, mouse immunization, and the various assays used to evaluate immune responses. This transparency allows for reproducibility and adds credibility to the findings. The results are presented clearly, with appropriate use of figures and tables to illustrate the findings. The statistical analyses are sound, and the data interpretation is consistent with the results obtained.

However, several aspects could be improved

1.      While the study effectively demonstrates the vaccine's immunogenicity and safety in mice, the manuscript acknowledges that further testing in target species, such as domestic dogs and wildlife, is necessary. The current study is limited in its ability to predict the vaccine's efficacy and safety across different species.

2.      The manuscript lacks data on the long-term immune responses elicited by the multiepitope vaccine. Understanding the duration of immunity and the potential need for booster doses is crucial for evaluating the vaccine's practical application.

3.      Although the study includes a commercial vaccine (CV) as a control, a more in-depth comparison with existing live attenuated or recombinant CDV vaccines could provide additional insights into the relative advantages or limitations of the new vaccine.

4.      The discussion highlights potential issues with the bioavailability and MHC presentation of the single peptide mixture, which did not elicit a significant immune response. Further exploration of strategies to enhance peptide presentation and stability could improve vaccine efficacy.

5.      The manuscript briefly mentions the ethical approval for the animal studies but could benefit from a more detailed discussion on the ethical implications of vaccine testing in wildlife species, particularly those that are endangered.

In conclusion, this manuscript presents a valuable contribution to the field of veterinary vaccine development, with innovative approaches and thorough experimentation. However, further research is needed to fully assess the vaccine's potential across different species and to address the limitations identified.

Author Response

  1. While the study effectively demonstrates the vaccine's immunogenicity and safety in mice, the manuscript acknowledges that further testing in target species, such as domestic dogs and wildlife, is necessary. The current study is limited in its ability to predict the vaccine's efficacy and safety across different species.

R/ we totally agree with reviewer. As mentioned at the end of the manuscript, conclusion section, further studies must be accomplished to demonstrated vaccine's efficacy and safety across different species.  However, we do believe that our approach proposed in the previous article (https://doi.org/10.1038/s41598-024-67781-5, citation 41), and evaluated in this paper, by using the consensus sequence  from all the linages reported for H and F proteins used to predict peptides could be a first step in the development of a safe interspecies vaccine. However, see manuscript changes highlighted in green in lines 498-500.

  1. The manuscript lacks data on the long-term immune responses elicited by the multiepitope vaccine. Understanding the duration of immunity and the potential need for booster doses is crucial for evaluating the vaccine's practical application.

R/ we agree with reviewer. The long-term immune response was not evaluated. Considering the immunogen type, booster are recommended and we have done 2 boosters based on the same facts. The importance of long-term immune response was clearly stated on perspectives. See manuscript changes highlighted in green in lines 442-444.

  1. Although the study includes a commercial vaccine (CV) as a control, a more in-depth comparison with existing live attenuated or recombinant CDV vaccines could provide additional insights into the relative advantages or limitations of the new vaccine.

R/ we partially agree with reviewer. We have employed a recombinant vaccine to compare the evaluated peptide-based vaccine. It is well known live attenuated vaccines have the advantage to induce humoral and cellular immune responses, as part of the diversification of the immune response. However, we could not do comparisons to other vaccines. The relevance of a new vaccine has arisen from the statement that has been established for other authors as Wilkes, Anis, among others, since different linages have emerged for which, current vaccines could not be protective (10.3390/pathogens12010057, 10.1016/j.vetmic.2018.03.014). Through the Discussion section we have selected the comparison not only with recombinant vaccines, but also with other current experimental vaccine candidates. This was clearly stated. See manuscript changes highlighted in green in lines 467-473.

  1. The discussion highlights potential issues with the bioavailability and MHC presentation of the single peptide mixture, which did not elicit a significant immune response. Further exploration of strategies to enhance peptide presentation and stability could improve vaccine efficacy.

R/ we agree with reviewer. We have mentioned limitations and possible strategies to improve it. Thus, this was clearly stated. See manuscript changes highlighted in green in lines 423-425.

  1. The manuscript briefly mentions the ethical approval for the animal studies but could benefit from a more detailed discussion on the ethical implications of vaccine testing in wildlife species, particularly those that are endangered.

R/ we totally agree with reviewer. Wilkes has proposed the importance of vaccination of wildlife animals (10.3390/pathogens12010057). We have also demonstrated in a scoping review (10.3390/v16071078) the lack of studies of wildlife animals vaccination trials, considering the limitations and challenges that must be faced to develop effective vaccines for wildlife animals, especially those in danger. Thus, this was clearly stated. See manuscript changes highlighted in green in lines 467-471.

In conclusion, this manuscript presents a valuable contribution to the field of veterinary vaccine development, with innovative approaches and thorough experimentation. However, further research is needed to fully assess the vaccine's potential across different species and to address the limitations identified.

R/ we agree with reviewer. Those facts have been stated in perspective section, however, this was clearly presented on perspectives. See manuscript changes highlighted in green in lines 467-471 and 498-500.

Reviewer 2 Report (Previous Reviewer 2)

Comments and Suggestions for Authors

Canine distemper is one of the most important infectious diseases affecting wild and domestic carnivores, which makes the topic of the manuscript important.

The introduction is well-written and provides enough information about the problem with canine distemper virus. However, there are the following recommendations, it is imperative to give more information about the vaccines being developed for CDV, in particular the recombinant CDV (rCDV) vaccine.

Further, when discussing multi-epitope vaccines, it is necessary to point out not only their advantages but also their disadvantages.

L127 Please, explain the composition of Recombitek c3 vaccine

L140 Explain GRIMACE score

L212 The optical density (OD) values were calculated by subtracting the negative control values from all samples.  indicate what kind of negative control you used

commercial vaccine (CV), indicate what type of vaccine is!

In the cited article, I did not find that they examined the levels of Th1/Th2/Th17 CD4+ cells, which makes the quote inaccurate.

My advice is that in the discussion you should not focus on the shortcomings of existing vaccines but on your contribution to uncovering the immune response to peptides and multiepitope polypeptides. 

The discussion needs to be rewritten, making it more focused on your results, and I would reduce the conclusion, and the information about the shortcomings of your study should be included in the discussion.

Author Response

Canine distemper is one of the most important infectious diseases affecting wild and domestic carnivores, which makes the topic of the manuscript important.

The introduction is well-written and provides enough information about the problem with canine distemper virus. However, there are the following recommendations, it is imperative to give more information about the vaccines being developed for CDV, in particular the recombinant CDV (rCDV) vaccine.

R/ we agree with reviewer. See manuscript changes highlighted in green in lines 66-70. See also citation 41 where we have published the design and evaluation of peptide-based vaccine (Rendon-Marin, S., Ruíz-Saenz, J. Universal peptide-based potential vaccine design against canine distemper virus (CDV) using a vaccinomic approach. Sci Rep 14, 16605 (2024). https://doi.org/10.1038/s41598-024-67781-5)

Further, when discussing multi-epitope vaccines, it is necessary to point out not only their advantages but also their disadvantages.

R/ we partially agree with reviewer. See manuscript changes highlighted in green in lines 438-445 to point out better the disadvantages.

L127 Please, explain the composition of Recombitek c3 vaccine

R/ we agree with reviewer. Recombitek c3 is a commercial vaccine employed as CDV vaccine as mentioned in methods section. Its composition is not clear, however, see manuscript changes highlighted in green in lines 134-136, to clarify the company.

L140 Explain GRIMACE score

R/ we agree with reviewer. See manuscript changes highlighted in green in lines 147-148.

L212 The optical density (OD) values were calculated by subtracting the negative control values from all samples.  indicate what kind of negative control you used

R/ we agree with reviewer. See manuscript changes highlighted in green in lines 222-223.

commercial vaccine (CV), indicate what type of vaccine is!

R/ we disagree with reviewer. In methods section the vaccine type was mentioned for commercial vaccine, Recombiteck c3. However, see manuscript changes highlighted in green in lines 134-136.

In the cited article, I did not find that they examined the levels of Th1/Th2/Th17 CD4+ cells, which makes the quote inaccurate.

R/ we partially agree with reviewer. The text was checked. See manuscript changes highlighted in green in lines 404-409.

My advice is that in the discussion you should not focus on the shortcomings of existing vaccines but on your contribution to uncovering the immune response to peptides and multiepitope polypeptides.

R/ we agree with reviewer. Different inclusions improved the discussion. See this section in the manuscript.

The discussion needs to be rewritten, making it more focused on your results, and I would reduce the conclusion, and the information about the shortcomings of your study should be included in the discussion.

R/ we agree with reviewer. Different inclusions improved the discussion.We were unable to shortened more, due to different request  from other reviewers. See this section in the manuscript.

Reviewer 3 Report (New Reviewer)

Comments and Suggestions for Authors

The study evaluated the safety and immune response of a new CDV multiepitope polypeptide vaccine in mice, showing it to be safe and capable of inducing significant seroconversion and cellular immune responses. Major concerns include 1) peptide designed may not work for dogs as MHC restrictions for presenting peptides, so immunogenicity testing in dogs is necessary; 2) Since both humoral and cellular responses are critical for controlling viral infection, the design did not consider protective nAb epitopes which are often conformational. Much more computational analysis is needed; 3) lack of detailed polypeptide information including epitope source, rationale, in vitro validation of expression, etc.

Author Response

The study evaluated the safety and immune response of a new CDV multiepitope polypeptide vaccine in mice, showing it to be safe and capable of inducing significant seroconversion and cellular immune responses.

Major concerns include

1) peptide designed may not work for dogs as MHC restrictions for presenting peptides, so immunogenicity testing in dogs is necessary;

R/ we agree with reviewer. Experiments on dogs must be carried out. However, this was clearly stated on perspectives. See manuscript changes highlighted in green in lines 498-500.

2) Since both humoral and cellular responses are critical for controlling viral infection, the design did not consider protective nAb epitopes which are often conformational. Much more computational analysis is needed;

R/ We partially agree with reviewer. We have published an article titled: Universal peptide-based potential vaccine design against canine distemper virus (CDV) using a vaccinomic approach (10.1038/s41598-024-67781-5). In this article, which is cited as a previous study, we reported the prediction and design of peptides from and in silico and in vitro approach. The prediction was made based on CD4+and CD8+ T cells, and linear B cells epitopes predicted with different computational tools. Please Check our previous paper (citation 41 in the manuscript), to see the design. (Rendon-Marin, S., Ruíz-Saenz, J. Universal peptide-based potential vaccine design against canine distemper virus (CDV) using a vaccinomic approach. Sci Rep 14, 16605 (2024). https://doi.org/10.1038/s41598-024-67781-5).

3) lack of detailed polypeptide information including epitope source, rationale, in vitro validation of expression, etc.

R/ we partially agree with reviewer. A mentioned, the peptide prediction, design and evaluation was reported in other publication. Similar to previous response, Please Check our article, to see the vaccine design (Rendon-Marin, S., Ruíz-Saenz, J. Universal peptide-based potential vaccine design against canine distemper virus (CDV) using a vaccinomic approach. Sci Rep 14, 16605 (2024). https://doi.org/10.1038/s41598-024-67781-5)

Round 2

Reviewer 2 Report (Previous Reviewer 2)

Comments and Suggestions for Authors

Overall, I believe that the article has significantly improved the quality of presentation of the research objectives and the results obtained. I would recommend that the conclusion be rewritten by synthesizing it in 3-5 sentences that show the main achievements of the research.

Author Response

Overall, I believe that the article has significantly improved the quality of presentation of the research objectives and the results obtained. I would recommend that the conclusion be rewritten by synthesizing it in 3-5 sentences that show the main achievements of the research.

R/. We totally agree to the reviewer. The conclusion was rewritted

Reviewer 3 Report (New Reviewer)

Comments and Suggestions for Authors

No comments

Author Response

No comments.

R/. Thanks for your review.

This manuscript is a resubmission of an earlier submission. The following is a list of the peer review reports and author responses from that submission.

Round 1

Reviewer 1 Report

Comments and Suggestions for Authors

Overall, this manuscript presents valuable findings that contribute to the field of CDV vaccine development. Its merits include 1) The study investigates the use of a multiepitope polypeptide vaccine for CDV, which is an innovative approach that combines computational biology and immunoinformatics with traditional experimental methods. This can potentially lead to the development of more effective vaccines. 2) The study employs a well-rounded approach, including in vivo experiments in mice, ELISA for seroconversion, flow cytometry for cellular immune responses, and cytokine quantification. This thorough approach ensures robust data supporting the conclusions. 3) The results indicating that the multiepitope CDV polypeptide vaccine is both safe and immunogenic are significant. This finding is crucial for advancing vaccine development for CDV, particularly given the challenges with current vaccines. 4) The manuscript is well-organized, and data are presented clearly in figures and tables, making it easy to follow the experimental results and understand the conclusions drawn.

However, addressing the following shortcomings and incorporating the suggested improvements will enhance the manuscript's impact and reliability.

a)     Some methodological details are lacking, such as the exact protocols used for splenocyte isolation and cytokine quantification. Including more detailed protocols would enhance reproducibility.

b)    The manuscript could benefit from a more in-depth discussion on the potential mechanisms by which the multiepitope polypeptide induces a stronger immune response compared to the single peptide mixture.

c)     The use of only six mice per group may limit the statistical power of the findings. Increasing the sample size could provide more robust data and strengthen the conclusions.

d)    While the study compares the multiepitope polypeptide vaccine with a commercial vaccine, a more detailed comparison with other vaccine types (e.g., live attenuated, subunit) would provide a broader context for the findings.

e)     The study focuses on short-term immune responses. Assessing the longevity of the immune response induced by the multiepitope polypeptide would be valuable for understanding its potential as a long-term vaccine solution.

Author Response

Reviewer 1.

Overall, this manuscript presents valuable findings that contribute to the field of CDV vaccine development. Its merits include 1) The study investigates the use of a multiepitope polypeptide vaccine for CDV, which is an innovative approach that combines computational biology and immunoinformatics with traditional experimental methods. This can potentially lead to the development of more effective vaccines. 2) The study employs a well-rounded approach, including in vivo experiments in mice, ELISA for seroconversion, flow cytometry for cellular immune responses, and cytokine quantification. This thorough approach ensures robust data supporting the conclusions. 3) The results indicating that the multiepitope CDV polypeptide vaccine is both safe and immunogenic are significant. This finding is crucial for advancing vaccine development for CDV, particularly given the challenges with current vaccines. 4) The manuscript is well-organized, and data are presented clearly in figures and tables, making it easy to follow the experimental results and understand the conclusions drawn.

However, addressing the following shortcomings and incorporating the suggested improvements will enhance the manuscript's impact and reliability.

  1. a) Some methodological details are lacking, such as the exact protocols used for splenocyte isolation and cytokine quantification. Including more detailed protocols would enhance reproducibility.

R/ We totally agree with reviewer. See the manuscript changes in those sections highlighted in yellow. CBA was carried out with according to manufacturer´s instructions.

  1. b) The manuscript could benefit from a more in-depth discussion on the potential mechanisms by which the multiepitope polypeptide induces a stronger immune response compared to the single peptide mixture.

R/ We totally agree with reviewer. See the manuscript changes highlighted in yellow in lines 403-414.

  1. c) The use of only six mice per group may limit the statistical power of the findings. Increasing the sample size could provide more robust data and strengthen the conclusions.

R/ We partially agree with reviewer. We have employed 6 mice per group since we needed 4 groups (Polypeptide, mixture commercial vaccine, and placebo)

Employing the following equation: n = t x r = 4 x 6 = 24

The interpretation is:

n – t = 24 - 4 = 20. Acceptable.

The number between 20-30 is acceptable, and considering the 3R, the minimum acceptable number was employed. 

doi: 10.4103/0976-500X.119726 (citation 37). See the manuscript changes highlighted in yellow in lines 137 and 138.

  1. d) While the study compares the multiepitope polypeptide vaccine with a commercial vaccine, a more detailed comparison with other vaccine types (e.g., live attenuated, subunit) would provide a broader context for the findings.

R/ We partially agree with reviewer. We have employed a widely used commercial vaccine, Recombitek. See the manuscript changes highlighted in yellow in lines 456-459, where we include this fact in the perspective section. Also, we have published a scoping review regarding CDV vaccination in domestic and wild animals (https://doi.org/10.3390/v16071078), where MVL and recombinant vaccines are the most common vaccines for CDV (citation 78)

  1. e) The study focuses on short-term immune responses. Assessing the longevity of the immune response induced by the multiepitope polypeptide would be valuable for understanding its potential as a long-term vaccine solution.

R/ We totally agree with reviewer. See the manuscript changes highlighted in yellow in lines 427-434, where we include this fact in the perspective section.

Reviewer 2 Report

Comments and Suggestions for Authors

Canine distemper virus severely threatens both domesticated and wild animals, including multiple carnivores, and the development of an effective vaccine is strongly desired. Various candidate CDV vaccines are being developed based on DNA vaccines, subunit vaccines, and vector vaccines. Rendon-Marin et al. offer an alternative to the existing CDV vaccine by testing the immunogenicity of peptides and multiepitope polypeptides.

Peptides and multiepitope polypeptides are characterized by low immunogenicity and weak induction of serum antibodies.  However, the proposed results do not support the idea that a multiepitope polypeptides-based vaccine can induce a protective immune response.

The introduction does not contain enough information about the structural proteins of CDV. Please provide information on neutralizing epitopes of CDV which are primarily presented on F and N proteins.

L58 MLV vaccines, explain the abbreviation and vaccine composition.

L76 Please, give a more detailed explanation of the provided examples. This sentence does not give a clear idea of ​​the specifics of multi-epitope-based vaccines (Chikungunya, Ebola, SARS-CoV-2, etc.).

Vaccines against multiple pathogens are normal based on chimeric VLPs, please give more information about VLPs as carriers of epitopes.

L 99 Hepatitis B vaccine is not peptide-based! You don’t have references to these examples.

In Table 1 you don’t have an explanation of CDV antigenic epitopes to which region they belong (F/N) and their aa position.

How did you determine which peptides to include in your study?

It is not clear what the final concentration of the vaccination dose is

L148 100 ml, probably it is a mistake

L193 CDV polypeptide – what is the composition of this polypeptide that you used as a coating Ag?

Why is Figure 2 showing data from 12 samples when you have 6 individuals in a group? If you are presenting data from biological replicates, I believe that arithmetic means from biological replicates should be presented. 

Author Response

Reviewer 2.

Canine distemper virus severely threatens both domesticated and wild animals, including multiple carnivores, and the development of an effective vaccine is strongly desired. Various candidate CDV vaccines are being developed based on DNA vaccines, subunit vaccines, and vector vaccines. Rendon-Marin et al. offer an alternative to the existing CDV vaccine by testing the immunogenicity of peptides and multiepitope polypeptides.

Peptides and multiepitope polypeptides are characterized by low immunogenicity and weak induction of serum antibodies. However, the proposed results do not support the idea that a multiepitope polypeptides-based vaccine can induce a protective immune response.

R/ We partially agree with reviewer. In lines 403-414, we have mentioned the limitations that must be overcome to improve peptide-based vaccines. In this study we have demonstrated the safety and immunogenicity of polypeptide, however, more studies must be accomplished to determine, with immunological challenge, the vaccine protection and usage of new adjuvants to activate innate immune system.

The introduction does not contain enough information about the structural proteins of CDV. Please provide information on neutralizing epitopes of CDV which are primarily presented on F and N proteins.

R/ We partially agree with reviewer. See the manuscript changes highlighted in yellow in lines 50-52 regarding H and F proteins, the main antigenic determinants and also, N protein.

L58 MLV vaccines, explain the abbreviation and vaccine composition.

R/ We totally agree with reviewer. See the manuscript changes highlighted in yellow in line 60.

L76 Please, give a more detailed explanation of the provided examples. This sentence does not give a clear idea of ​​the specifics of multi-epitope-based vaccines (Chikungunya, Ebola, SARS-CoV-2, etc.).

R/ We totally agree with reviewer. See the manuscript changes highlighted in yellow in lines 88-91 and summary was done.

Vaccines against multiple pathogens are normal based on chimeric VLPs, please give more information about VLPs as carriers of epitopes.

R/ We totally agree with reviewer. See the manuscript changes highlighted in yellow in lines 102-109.

L 99 Hepatitis B vaccine is not peptide-based! You don’t have references to these examples.

R/ We totally agree with reviewer. See the manuscript changes highlighted in yellow in lines 92-96 to clarify since hepatitis B cited alternative is a virus like particle carrying multiple epitopes.

In Table 1 you don’t have an explanation of CDV antigenic epitopes to which region they belong (F/N) and their aa position.

R/ We totally agree with reviewer. When the manuscript was submitted to this journal, the in silico and in vitro evaluation article was un publication process. Now, the paper has been published in Scientific Reports (doi: 10.1038/s41598-024-67781-5), citation 36. In this article, citation  in the new manuscript, has been cited. However, see the manuscript in Table 1 highlighted in yellow the new columns.

How did you determine which peptides to include in your study?

R/ We agree with reviewer. However, please check the new citation 36. We have published an article where we predicted and evaluated the employed peptides (doi: 10.1038/s41598-024-67781-5). We selected five that overcame the in silico and in vitro safety evaluation.

It is not clear what the final concentration of the vaccination dose is

R/ We totally agree with reviewer. See the manuscript changes highlighted in yellow in lines 150-152.

L148 100 ml, probably it is a mistake

R/ We totally agree with reviewer. See the manuscript changes highlighted in yellow in line 154.

L193 CDV polypeptide – what is the composition of this polypeptide that you used as a coating Ag?

R/ We totally agree with reviewer. The composition is the same as polypeptide employed as immunogen. See the manuscript changes highlighted in yellow in lines 200 and 201 to clarify.

Why is Figure 2 showing data from 12 samples when you have 6 individuals in a group? If you are presenting data from biological replicates, I believe that arithmetic means from biological replicates should be presented.

R/ We totally agree with reviewer. See Figure 2 presented as suggested and figure legend was modified and highlighted in yellow.

Reviewer 3 Report

Comments and Suggestions for Authors

GENERAL COMMENTS

In the study vaccines-3101966 entitled "Evaluation of the Safety and Immunogenicity of a Multiple Epitope Polypeptide from Canine Distemper Virus (CDV) in Mice," the authors evaluated a new generation vaccine based on CDV peptides in mice. The findings indicated that the multiepitope CDV polypeptide induced significant seroconversion and a stronger cellular immune response compared to the placebo, representing a significant advancement in CDV vaccine development. The manuscript presents compelling data on a topic of significant interest and is suitable for the journal. The English language is acceptable, and the manuscript's presentation and length are appropriate, including a clear experimental plan. The title accurately reflects the main findings, the keywords are representative of the article, and the abstract effectively summarizes the study's background, methodology, results, and significance. While the introduction is well-written, it could be condensed. The materials and methods section is thorough, but punctuation errors need correction. The results section is clear and well-explained. However, the discussion and conclusion sections are unclear, with overly long and convoluted paragraphs that require rewriting. Therefore, the manuscript is suitable for publication after major revisions.

SPECIFIC COMMENTS

  1. The title accurately reflects the main findings of the work.
  2. The abstract clearly summarizes the background, methodology, results, and significance of the study, but some sentences should be improved.
  3. The introduction is well-written but could be more concise.
  4. The materials and methods section is clear but needs punctuation corrections.
  5. The discussion and conclusion sections should be rewritten for clarity.
  6. Tables and figures are generally good and represent the results well, without duplicating data in the text.
  7. The reference list needs improvement. Ensure references are corrected and formatted according to the journal's style.

Author Response

Reviewer 3.

In the study vaccines-3101966 entitled "Evaluation of the Safety and Immunogenicity of a Multiple Epitope Polypeptide from Canine Distemper Virus (CDV) in Mice," the authors evaluated a new generation vaccine based on CDV peptides in mice. The findings indicated that the multiepitope CDV polypeptide induced significant seroconversion and a stronger cellular immune response compared to the placebo, representing a significant advancement in CDV vaccine development. The manuscript presents compelling data on a topic of significant interest and is suitable for the journal. The English language is acceptable, and the manuscript's presentation and length are appropriate, including a clear experimental plan. The title accurately reflects the main findings, the keywords are representative of the article, and the abstract effectively summarizes the study's background, methodology, results, and significance. While the introduction is well-written, it could be condensed. The materials and methods section is thorough, but punctuation errors need correction. The results section is clear and well-explained. However, the discussion and conclusion sections are unclear, with overly long and convoluted paragraphs that require rewriting. Therefore, the manuscript is suitable for publication after major revisions.

SPECIFIC COMMENTS

The title accurately reflects the main findings of the work.

R/ We totally agree with reviewer.

The abstract clearly summarizes the background, methodology, results, and significance of the study, but some sentences should be improved.

R/ We totally agree with reviewer. See the manuscript changes in the abstract highlighted in yellow to improve the writing.

The introduction is well-written but could be more concise.

R/ We totally agree with reviewer. See the manuscript changes to improve the introduction section.

The materials and methods section is clear but needs punctuation corrections.

R/ We totally agree with reviewer. See the manuscript changes highlighted in yellow in methods section.

The discussion and conclusion sections should be rewritten for clarity.

R/ We totally agree with reviewer. See the manuscript changes in those manuscript sections for more clarity.

Tables and figures are generally good and represent the results well, without duplicating data in the text.

R/ We totally agree with reviewer.

The reference list needs improvement. Ensure references are corrected and formatted according to the journal's style.

R/ We totally agree with reviewer. Changes in the citation style were done and they are highlighted in yellow.